# Rhythm Control and Cardiovascular or Cerebrovascular Outcomes in Patients with Atrial Fibrillation: A Study of the CODE-AF Registry

**DOI:** 10.3390/jcm12144579

**Published:** 2023-07-10

**Authors:** Ho-Gi Chung, Junbeom Park, Jin-Kyu Park, Ki-Woon Kang, Jaemin Shim, Jin-Bae Kim, Jun Kim, Eue-Keun Choi, Hyung Wook Park, Young Soo Lee, Boyoung Joung

**Affiliations:** 1Division of Cardiology, Department of Internal Medicine, Yonsei University College of Medicine, 50-1 Yonseiro, Seodaemun-gu, Seoul 03722, Republic of Korea; chung4579@yuhs.ac; 2Department of Cardiology, School of Medicine, Ewha Womans University, Seoul 07985, Republic of Korea; 3Department of Cardiology, Hanyang University Seoul Hospital, Seoul 04763, Republic of Korea; 4Division of Cardiology, Eulji University Hospital, Daejeon 35233, Republic of Korea; 5Division of Cardiology, Department of Internal Medicine, Korea University Medical Center, Seoul 02841, Republic of Korea; 6Division of Cardiology, Department of Internal Medicine, Kyung Hee University Hospital, Seoul 02447, Republic of Korea; 7Heart Institute, Asan Medical Center, University of Ulsan College of Medicine, Seoul 05505, Republic of Korea; 8Department of Internal Medicine, Seoul National University Hospital, Seoul 03080, Republic of Korea; 9Department of Cardiology, School of Medicine, Chonnam National University, Gwangju 61469, Republic of Korea; 10Division of Cardiology, Department of Internal Medicine, Daegu Catholic University Medical Center, Daegu 42472, Republic of Korea

**Keywords:** rhythm control, atrial fibrillation, oral anticoagulation, prognosis

## Abstract

Background: It is not clear whether the data regarding rhythm control during atrial fibrillation (AF) contained in AF registries is prognostically significant. Thus, this study investigated the relationship between rhythm control and cardiovascular outcomes in patients in contemporary AF registries. Methods: This study was conducted using data from 6670 patients with AF receiving oral anticoagulation in the CODE-AF registry. We used propensity overlap weighting to account for differences in baseline characteristics between the rhythm control and rate control groups. The primary outcome was a composite of the rate of death due to cardiovascular causes, stroke, acute coronary syndrome, and heart failure. The secondary outcomes were individual components of the primary outcome. Results: In the CODE-AF registry, 5407 (81.1%) patients were enrolled three months after AF diagnosis. During a median follow-up period of 973 days (interquartile range: 755–1089 days), a primary outcome event occurred in 72 patients in the rhythm control group (1.4 events per 100 person-years) and in 211 patients in the rate control group (1.8 events per 100 person-years). However, after overlap weighting, the incidence rates were 1.4 and 1.5 events per 100 person-years, respectively. No significant difference was found in either the primary outcome (weighted HR: 0.87; 95% CI: 0.66–1.17; *p* = 0.363) or secondary outcomes between the rhythm control and rate control groups. Conclusion: In a prospective AF registry in which most of the population was enrolled at least three months after AF diagnosis, no difference in the risk of cardiovascular or cerebrovascular outcomes was found between the rhythm control and rate control groups, suggesting the early rhythm control should be considered to improve the outcome of patients.

## 1. Introduction

Atrial fibrillation (AF) is associated with an increased risk of morbidity and mortality from congestive heart failure (HF) and stroke and an impaired quality of life, even for patients who receive optimal anticoagulation and rate control treatment [1,2,3,4]. Rate control is an integral part of managing AF and often sufficiently improves associated symptoms [1,2]. By restoring and maintaining sinus rhythm using antiarrhythmic drug treatment, cardioversion, and AF ablation, rhythm control improves patients’ symptoms and quality of life [5]. Several randomized trials, including the landmark Atrial Fibrillation Follow-up Investigation of Sinus Rhythm Management (AFFIRM) trial, have found no significant differences between rhythm and rate control in terms of their effects on mortality and stroke [6,7,8]. The Early Treatment of AF for Stroke Prevention Trial (EAST-AFNET 4) recently demonstrated that rhythm control was associated with a lower risk of adverse cardiovascular outcomes than traditional forms of care among patients diagnosed with AF within the preceding year [9].

Our previous study also showed that early initiation of rhythm control treatment was associated with a lower risk of adverse cardiovascular outcomes than rate control in patients with recently diagnosed AF. This association was not found in patients who had had AF for more than nine months [10]. Moreover, the strength of the positive effect of early rhythm control on cardiovascular outcomes decreases with age with larger benefits in patients <75 years of age [11]. Therefore, we cannot ignore the significant reduction of the event rate associated with early rhythm control in relatively young and low-risk patients ineligible for the EAST-AFNET 4 trial [12]. Moreover, early rhythm control is associated with a lower risk of dementia than rate control [13].

However, the cardiovascular and cerebrovascular effects of rhythm control therapy among people in AF registries are not well known. Yang et al. reported that AFFIRM subjects diagnosed with AF within six months of study enrollment showed no difference in survival, cardiovascular hospitalization, or ischemic stroke rates according to whether they received rate or rhythm control [14]. The superiority of rhythm control strategies reported by newer AF trials may be the product of AF therapy development, not intervention timing [14]. Therefore, this study was conducted to evaluate the effect of rhythm control therapy in AF patients by analyzing subjects from the contemporary COmparison study of Drugs for symptom control and complication prEvention of Atrial Fibrillation (CODE-AF) study.

## 2. Materials and Methods

### 2.1. Data Source and Study Population

The CODE-AF registry is an ongoing prospective observational registry at 18 tertiary hospitals from all geographical regions of South Korea. Detailed descriptions of the registry are available in previous studies [15,16,17,18]. In brief, the CODE-AF registry records the clinical epidemiology of patients with AF, the diagnostic and therapeutic processes they underwent, and their clinical outcomes. The CODE-AF registry was designed by the Korean Heart Rhythm Society. Data were accumulated in an electronic database to reduce inconsistencies and errors. All patients provided informed consent before registration. The CODE-AF registry was approved by the research ethics committee in each center (4-2016-0105) and registered at ClinicalTrials.gov, accessed on 30 May 2016 (NCT02786095).

A total of 12,953 AF patients who sought treatment between June 2016 and July 2020 were included in this study. To be included in the registry, patients had to be >18 years old, have non-transient AF with a reversible cause or need for chronic anticoagulation to treat other conditions, such as implanted prosthetic valves, deep vein thrombosis, or pulmonary thromboembolism. After enrollment in the CODE-AF registry, patients had a six-month follow-up either by telephone or outpatient clinic visit. Each patient’s demographics, detailed medical history, and laboratory data were recorded at enrollment.

AF was classified as paroxysmal, persistent, or permanent. Patients with AF terminated within 7 days of onset were defined as paroxysmal atrial fibrillation, whereas patients with AF sustained beyond 7 days of onset were defined as persistent AF, and permanent AF as persistent AF for more than 12 months. HF diagnosis was based on medical history or clinical judgment. Patients with HF and left ventricular ejection fraction (LVEF) ≥ 40% were defined as HF with preserved ejection fraction (HFpEF), whereas patients with HF and an LVEF < 40% were defined as HF with reduced ejection fraction (HFrEF). Patients who did not make an outpatient clinic visit, were not in telephone contact for more than 365 days, for whom baseline medical data or echocardiography measures were not available, or who were permanently paced were excluded from our study sample.

To remove the effect of anticoagulation, 7767 patients taking oral anticoagulation (OAC) were included from our study sample. Additionally, 1097 patients for whom there was no follow-up were excluded, leaving 6670 patients remaining. These patients were categorized into either the rhythm control group or rate control group according to whether they had an antiarrhythmic drug consumption or AF ablation history.

### 2.2. Rate and Rhythm Control Strategies

Digoxin, beta-blockers, or both were used for rate control in patients with AF and HFrEF. In patients with AF with an LVEF ≥ 40%, beta-blockers, digoxin, and non-DHP calcium channel blockers, such as verapamil or diltiazem, were used.

In patients with normal left ventricle function without pathological left ventricular hypertrophy, dronedarone, flecainide, propafenone, or sotalol was used for recurrent symptomatic AF. Dronedarone, sotalol, or amiodarone is recommended for recurrent symptomatic AF in patients with a history of coronary artery disease without HF. Amiodarone is recommended for patients with HF with recurrent symptomatic AF. AF ablation can be used in all AF patients.

### 2.3. Outcome Definition and Follow-Up

The primary outcome was a composite of death from cardiovascular causes, stroke, acute coronary syndrome, and HF requiring hospitalization. The secondary outcomes were the individual components of the primary outcome. Stroke was defined as cerebral, spinal, or retinal infarction causing neurological dysfunction without using the NIH stroke scale. Myocardial infarction was defined as the rise and/or fall of troponin T at least one value above the 99th percentile and with at least one of the following: symptoms of ischemia, pathologic Q waves in the electrocardiogram, new significant ST-T wave changes or new onset LBBB, intracoronary thrombus by angiography or autopsy, new regional wall motion abnormality, or new loss of viable myocardium by echocardiography [19]. HF requiring hospitalization was defined as hospital admission due to new onset or worsening signs and symptoms of HF irrespective of LVEF. Patients were followed up with until the study outcome occurred, death, the study period ended, or censoring was deemed necessary.

### 2.4. Statistical Analysis

The baseline characteristics of the rhythm control and rate control groups were compared and summarized as mean and standard deviation for continuous variables and counts and percentages for categorical variables. We used the propensity overlap weighting method to account for differences in the baseline characteristics between the rhythm control and rate control groups. The propensity score, the probability of receiving rhythm control, were estimated using a general logistic regression model based on demographics, including age and sex, vital signs, BMI, type of AF, date of AF onset, CHA_2_DS_2_-VASc and HAS BLED clinical risk scores, past medical history, and concurrent rhythm control and rate control medication (Table 1). The overlap weight for patients receiving rhythm control and rate control was defined as 1 minus the propensity score and the propensity score, respectively. The overlap weight was used to estimate the population-average treatment effects while minimizing treatment effect variance. The balance between the rhythm control and rate control strategy groups was evaluated using a threshold of a standardized mean difference of 0.1 to all baseline covariates to indicate imbalance.

Time-to-events was defined as the number of days from the date of entry into the CODE-AF registry to the date of the medical event as a study endpoint. The weighted incidence rate was calculated by dividing the weighted number of clinical events during the follow-up period by 100 person-years at risk. Survival free of the primary outcome is presented with the Kaplan-Meier (KM) method and compared using the weighted log-rank test.

Hazard ratios (HR) and 95% confidence intervals (95% CI) from a Cox regression model fitted to new weights were calculated using the rate control group as the reference category for both primary and secondary outcomes to show the associations between rhythm control and cardiovascular or cerebrovascular outcomes. A univariable Cox regression model fitted to new weights was used to identify variables associated with outcome development.

*p*-values lower than 0.05 were considered statistically significant. Statistical analysis was conducted using R software version 4.2.2 (R Foundation for Statistical Computing, Vienna, Austria).

### 2.5. Subgroup Analysis

We performed subgroup analysis for the primary composite outcome stratified by sex, age (age < 75 versus age ≥ 75), the onset of AF (enrollment in the registry within three months of diagnosis versus enrollment after three months), CHA_2_DS_2_-VASc score, previous stroke history, and HF history. Baseline characteristic variables were used to create overlap weights by propensity overlap weighting. A Cox proportional hazard model was fitted to new weights and interaction tests were conducted for all subgroups. Subgroup analysis according to AF onset was included because the impact of rhythm control on cardiovascular outcomes can vary by AF onset.

## 3. Results

### 3.1. Study Population

The baseline characteristics of the study population in the CODE-AF registry are presented in Table 1. Among the 6670 patients with AF and OAC (male: 4152, mean age [SD]: 69 [11]), 4506 (67.6%) patients and 2164 (32.4%) patients received rate and rhythm control therapy, respectively. Patients in the rhythm control group were less likely to have comorbidities, such as hypertension, diabetes, dyslipidemia, chronic kidney disease (CKD), and stroke; tended to be younger; tended to have lower CHA_2_DS_2_-VASc scores; were more likely to use warfarin and antiplatelets; and less likely to use NOACs, beta-blockers, calcium channel blockers, digitalis, and ACEi/ARB than those in the rate control group. Only 1263 (18.9%) patients were enrolled within three months of AF diagnosis, while 5407 (81.1%) were enrolled after three months from AF diagnosis.

The most commonly used rhythm control drugs were the class Ic drug flecainide, which was used by 667 patients (30.8%), propafenone, which was used by 591 patients (27.3%), and the class III drug amiodarone (26.4%), which was used by 571 patients. Ablation was the initial rhythm control strategy for 141 patients (6.5%) and was performed during follow-up in 309 patients (14.3%) in the rhythm control group (Figure 1). The patients with ablation were relatively younger with fewer comorbidities (Appendix A).

After overlap weighting, all baseline characteristics were similar between the rhythm control and rate control groups (Table 1).

### 3.2. Clinical Outcomes

During a median follow-up period of 973 days (interquartile range: 755–1089 days), a primary outcome event occurred in 72 of the patients in the rhythm control group (1.4 events per 100 person-years) and in 211 patients assigned to rate control group (1.8 events per 100 person-years) (Table 2). Before weighting, there was a lower cumulative incidence of the primary outcome (log rank *p* = 0.022) (Appendix A) and a lower risk of the primary outcome in the rhythm control group than in the rate control group (HR: 0.73, 95% CI: 0.59–0.95) (Table 2). However, after overlap weighting, the incidence of the primary outcome was 1.4 and 1.5 events per 100 person-years for the rhythm control and rate control groups, respectively. There was no statistically significant difference in the risk of the primary outcome between the rhythm control and rate control strategy groups (weighted HR: 0.87, 95% CI: 0.66–1.17, *p* = 0.363) (Table 3). The cumulative incidence curve of the weighted primary outcome is presented in Figure 2.

Among secondary outcomes, 101 patients experienced a stroke event of which 20 were in the rhythm control group and 81 were in the rate control group at rates of 0.4 and 0.7 events per 100 person-years, respectively (Table 2). Before weighting, the rhythm control group had a lower cumulative incidence (log rank *p* = 0.010) (Appendix A) and a lower risk of stroke than the rate control group (HR: 0.52, 95% CI: 0.32–0.85) (Table 2). However, there was no difference in the rhythm and rate control groups’ weighted incidence and weighted HR, with 0.4 and 0.6 events per 100 person-years, respectively (weighted HR: 0.60, 95% CI: 0.36–1.02, *p* = 0.060) (Table 3).

Among other secondary outcomes, 25 patients experienced cardiovascular (CV) death (weighted incidence of 0.1 vs. 0.1 per 100 person-years), 26 experienced acute coronary syndrome (weighted incidence of 0.1 vs. 0.2 per 100 person-years), and 23 were admitted with worsening heart failure (weighted incidence of 0.1 vs. 0.1 per 100 person-years) (Table 2 and Table 3). There was no difference in the weighted HRs of secondary outcomes between the two groups (Table 3). The cumulative incidence curves of weighted secondary outcomes are presented in Figure 3.

### 3.3. Subgroup Analysis

The primary outcome for the rhythm control group compared with those of the rate control group stratified by subgroups of interest are presented before weighting in Appendix A. After weighting, the effect of rhythm and rate control on the primary outcome was the same and did not vary by sex, age, the onset of AF, CHA_2_DS_2_-VASc score, previous stroke history, or HF history. There was no association between subgroups of interest and the primary outcome (Table 4).

## 4. Discussion

In this contemporary prospective multicenter cohort study, rhythm control produced no difference in the primary outcome, which included cardiovascular death, stroke, acute coronary syndrome, or hospitalization due to heart failure, from rate control. Stratified analysis of CODE-AF participants with AF diagnosed within three months of enrollment did not reveal any significant clinical benefit of rhythm control over rate control in an intention-to-treat analysis. This finding suggests that initiating rhythm control therapy as soon as possible after AF diagnosis is important.

### 4.1. The Timing of Rhythm Control Initiation

Before balancing, rhythm control was associated with a reduced incidence and risk of the primary outcome and stroke, which was consistent with our previous study [17]. However, after weighing, there was no difference in the effects of rhythm or rate control. This result can be explained by the fact that a low proportion of the registry’s population belonged to the early AF rhythm control group. Considering that only 1263 (18.9%) of the patients enrolled in the CODE-AF registry within three months of AF diagnosis, rhythm control initiation might be further delayed. Among patients enrolled in the CODE-AF registry within three months of AF diagnosis, defining an enlarged left atrium as a left atrial diameter of more than 4.0 cm, 69.7% of these patients had an enlarged left atrium. This result means that the number of patients who met the EAST-AFNET 4 trial’s definition of early AF was small [9]. Indeed, the median duration from AF diagnosis to AF initiation was one month in the EAST-AFNET 4 trial. Furthermore, 39.1% of the CODE-AF registry rhythm control participants had persistent or permanent AF. This study likely produced a different result from the EAST-AFNET 4 trial because only 26% of EAST-AFNET 4 trial participants had persistent AF [9]. The rhythm control effect was more refractory in patients with persistent or permanent AF than paroxysmal AF. A similar result was observed in the subgroup analysis of the AFFIRM study [14].

AF management is limited by its low detection rate. Atrial cardiomyopathy can be largely reversible, but as it progresses, time for detection is lost and the disease’s manifestations will become less reversible [20]. Early detection of AF and rhythm control by catheter ablation or cryoablation will have better outcomes than those who received regular medical therapy by intention-to-treat analysis. [14,21,22] Recently, implanted devices, wearables, and other consumer electronics that enable long-term continuous monitoring of electrocardiograms (ECG) have been developed, allowing for the earlier detection of AF [20]. Mobile technologies have been demonstrated to help reduce the rates of stroke, thromboembolism, all-cause death, and rehospitalization among AF patients compared to usual care [23]. The screening population, method of screening, and analysis timing should be defined with mobile health technology to reduce the cost and increase the effectiveness of AF screening and to identify candidates for early rhythm control, as described in EAST-AFNET 4 Trial [24,25]. However, substantial challenges to achieving this goal remain, including government laws regarding mobile device data protection, the fact that elderly users are less able to use mobile devices, device reliability, and the privacy of data when it is stored in databases [20]. Reduction of the burden of AF complications by early diagnosis through systematic screening by rhythm control seems possible if the above problems are resolved [26].

### 4.2. The Study Population

Differing baseline characteristics between this study’s patient population and that of the EAST-AFNET 4 could also have produced different results in these studies. Participants in the EAST-AFNET 4 trial were more likely to have high CHA_2_DS_2_-VASc scores and many comorbidities, such as hypertension, congestive heart failure, and chronic kidney disease, than those in the CODE-AF registry. A subgroup analysis of the EAST-AFNET 4 trial showed that patients with recently diagnosed AF and CHA_2_DS_2_-VASc scores ≥4 had a lower risk of a primary outcome of death from cardiovascular causes, stroke, and hospitalization for the worsening of heart failure or acute coronary syndrome than those with lower scores and those with fewer comorbidities had less favorable outcomes with early rhythm control than rate control [27]. The average CHA_2_DS_2_-VASc score of CODE-AF registry participants is 3.0. Moreover, the prevalence of heart failure was lower among patients in the CODE-AF registry (rhythm control: 12.1%, rate control: 11.6%) than those in the EAST-AFNET 4 trial (rhythm control: 28.4%, rate control: 28.8%). In the early rhythm control therapy in patients with AF and heart failure subgroup analysis of the EAST-AFNET 4 trial who were defined as having heart failure symptoms of New York Heart Association II to III or an LVEF <50%, the rhythm control strategy group had lower rates of primary outcomes of cardiovascular death, stroke, or hospitalization for the worsening of heart failure or acute coronary syndrome than the rate control group [28]. Based on these studies, there would not likely be any difference in cardiovascular or cerebrovascular outcomes between the rhythm control group and the rate control group.

### 4.3. Rhythm Control Methods

Two studies, including the AFFIRM study, conducted two decades ago, did not find a difference between the rhythm control and rate control groups, but many have hypothesized that modern rhythm control methods could produce different results [6,7]. If more modern rhythm control therapy was used in the EAST-AFNET 4 trial that is superior to other antiarrhythmic drugs, the result might have been different. In this study, catheter ablation was used as an initial therapy in 8% of patients in the rhythm control group and in 19.4% of patients at 2 years follow-up in the EAST-AFNET 4 trial. The Catheter Ablation versus Antiarrhythmic Drug Therapy in Atrial Fibrillation randomized trial showed that patients who received catheter ablation had better outcomes than those who received regular medical therapy by intention-to-treat analysis [14]. In the CODE-AF registry analysis, therapy was initially given to 6.5% of patients in the rhythm control and during follow-up to 14.3% of patients in the same group.

The results of this secondary analysis are important for clinicians. The fact that the results of the EAST-AFNET 4 trial could not be corroborated by the new AF subgroup of the AFFIRM trial does raise the question of whether the benefit shown in the EAST-AFNET 4 trial was due to early rhythm control or another factor in the intervention arm of the study. For example, those patients assigned to the early rhythm control group had a follow-up that included electrocardiogram transmission twice per week in the event of symptoms, while the control group did not [9]. This treatment may have increased treatment compliance in the intervention arm. Indeed, providing AF patients with a structured follow-up by using mobile technologies has recently been demonstrated [23].

### 4.4. Limitations

First, since the CODE-AF registry is not a randomized control study, different baseline demographic characteristics between the rate and rhythm control groups were observed. However, after overlap weighting, all of the groups’ baseline covariates were similar. Second, we divided patients according to whether they were diagnosed with AF within or after three months of their enrollment in the CODE-AF registry. Therefore, this does not reflect when rhythm or rate control therapy was initiated. Third, the follow-up period of the CODE-AF registry is relatively short. Patients stopped participating in the EAST-AFNET 4 trial after a median of 5.1 years compared to 2.7 years for those enrolled in the CODE-AF registry. The difference in median follow-up may limit our ability to detect statistically meaningful differences in the cumulative incidence and hazard ratios between the rhythm control and rate control groups.

## 5. Conclusions

There was no difference between AF patients who received oral anticoagulants according to whether they enrolled in the CODE-AF registry within or after three months following their AF diagnosis and according to whether they received rate or rhythm control in terms of death from cardiovascular causes, stroke, hospitalization due to worsening of HF, and acute coronary syndrome. This result suggests that rhythm control therapy should be initiated upon AF diagnosis. Detecting AF early will help better identify early rhythm control candidates as defined in the EAST-AFNET 4 trial.

## Figures and Tables

**Figure 1 jcm-12-04579-f001:**
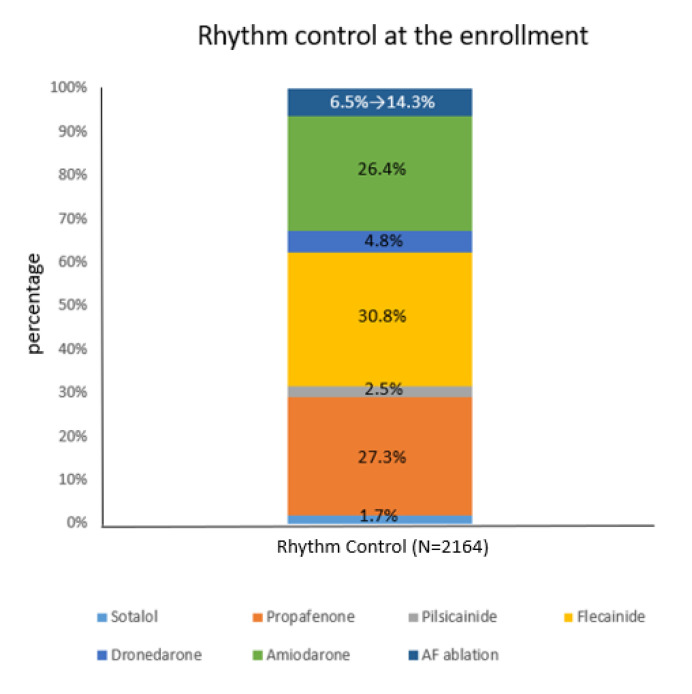
The most commonly used rhythm control drugs were the class Ic drug flecainide, which was used by 667 patients (30.8%), propafenone, which was used by 591 patients (27.3%), and the class III drug amiodarone (26.4%), which was used by 571 patients. Ablation was the initial rhythm control strategy for 141 patients (6.5%) and was performed during follow-up in 309 patients (14.3%) in the rhythm control group.

**Figure 2 jcm-12-04579-f002:**
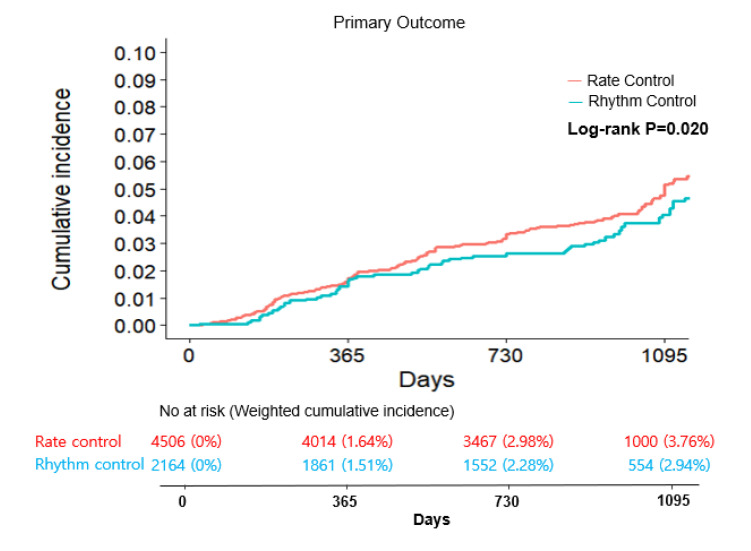
Weighted cumulative incidence curve for the primary composite outcome in rhythm control and rate control groups for atrial fibrillation.

**Figure 3 jcm-12-04579-f003:**
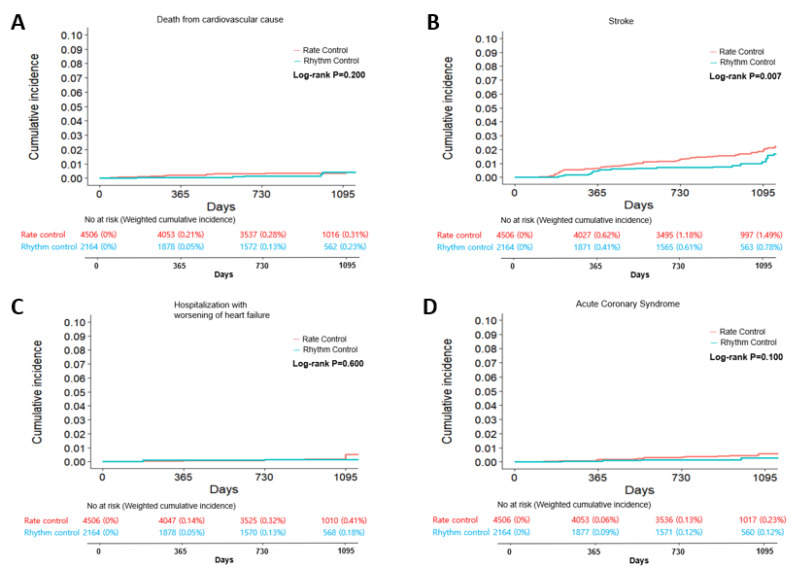
Weighted cumulative incidence curves for individual components of the primary composite outcome in rhythm control and rate control groups for atrial fibrillation. Death from cardiovascular cause (**A**), stroke (**B**), hospitalization with worsening of heart failure (**C**), and acute coronary syndrome (**D**).

**Table 1 jcm-12-04579-t001:** Baseline demographic characteristics before and after overlap weighting.

	Before Overlap Weighting			After Overlap Weighting		
	Rate Control (N = 4506)	Rhythm Control (N = 2164)	ASD	Rate Control (N = 1293 [4506] *)	Rhythm Control (N = 1293 [2164] *)	ASD
Age in years	70.1 (9.7)	66.0 (12.1)	0.379	67.6 (10.3)	67.6 (12.8)	<0.001
Male	2712 (60.2)	1440 (66.5)	0.132	829 (64.1)	829 (64.1)	<0.001
Body mass index, kg/m^2^	24.8 (3.5)	24.9 (3.2)	0.032	24.9 (3.4)	24.9 (3.2)	<0.001
Systolic BP, mmHg	122.3 (15.8)	124.7 (15.4)	0.156	124.0 (16.0)	124.0 (15.3)	<0.001
Diastolic BP, mmHg	74.3 (12.0)	75.6 (17.1)	0.090	74.9 (12.0)	74.9 (11.7)	<0.001
Heart rate, beats/min	77.4 (21.4)	75.6 (17.1)	0.091	75.9 (18.1)	75.9 (17.5)	<0.001
Type of AF			0.228			<0.001
Paroxysmal	2649 (58.8)	1319 (61.0)		796 (61.5)	796 (61.5)	
Persistent	1634 (36.3)	822 (38.0)		480 (37.1)	480 (37.1)	
Permanent	222 (4.9)	23 (1.1)		18 (1.4)	18 (1.4)	
Onset of AF			0.052			<0.001
<3 month	830 (18.4)	433 (20.0)		253 (19.6)	253 (19.6)	
≥3 month	3676 (81.6)	1731 (80.0)		1040 (80.4)	1040 (80.4)	
Alcohol intake **	1197 (26.6)	646 (29.8)	0.082	371 (28.7)	371 (28.7)	<0.001
Current smoking	1270 (28.2)	716 (33.1)	0.107	402 (31.0)	401 (31.0)	<0.001
CHA_2_DS_2_-VASc score	3.2 (1.6)	2.6 (1.6)	0.382	2.8 (1.5)	2.8 (1.6)	<0.001
HAS BLED Score ***	2.0 (1.0)	1.7 (1.1)	0.282	1.8 (1.0)	1.8 (1.0)	<0.001
Hypertension	3315 (73.7)	1405 (65.2)	0.185	885 (68.4)	885 (68.4)	<0.001
Diabetes	1417 (31.5)	560 (26.0)	0.122	359 (27.7)	359 (27.7)	<0.001
Dyslipidemia	1663 (36.9)	715 (33.0)	0.081	445 (34.4)	445 (34.4)	<0.001
Myocardial infarction	121 (2.7)	77 (3.6)	0.069	40 (3.1)	40 (3.1)	<0.001
Congestive heart failure	542 (12.1)	249 (11.6)	0.018	147 (11.4)	147 (11.4)	<0.001
Peripheral vascular disease	253 (5.6)	128 (5.9)	0.013	72 (5.6)	72 (5.6)	<0.001
Stroke	896 (19.9)	335 (15.5)	0.116	225 (17.4)	225 (17.4)	<0.001
CKD ****	506 (11.2)	191 (8.8)	0.080	121 (9.4)	121 (9.4)	<0.001
Medications						
NOAC	3530 (78.3)	1630 (75.3)	0.072	998 (77.1)	998 (77.1)	<0.001
Warfarin	1076 (23.9)	642 (29.7)	0.131	345 (26.7)	345 (26.7)	<0.001
Antiplatelet	391 (8.7)	307 (14.2)	0.174	152 (11.7)	152 (11.7)	<0.001
Beta-blocker	2446 (54.3)	1027 (47.5)	0.136	639 (49.4)	639 (49.4)	<0.001
CCB	1360 (30.2)	552 (25.5)	0.104	351 (27.1)	351 (27.1)	<0.001
Digitalis	459 (10.2)	65 (3.0)	0.292	52 (4.0)	52 (4.0)	<0.001
Diuretics	362 (8.1)	157 (7.3)	0.030	97 (7.5)	97 (7.5)	<0.001
ACEi/ARB	2015 (44.7)	891 (41.2)	0.071	543 (42.0)	543 (42.0)	<0.001
Statin	1702 (37.8)	804 (37.2)	0.012	484 (37.5)	484 (37.5)	<0.001

Values are presented as mean (standard deviation) or number (%). * Weighted number of individuals [crude number of individuals]. ** Social drinking and drinking were considered as alcohol intake. *** Modified HAS-BLED = hypertension, 1 point: >65 years old, 1 point: stroke history, 1 point: bleeding history or predisposition, 1 point: liable international normalized ratio, not assessed: ethanol or drug abuse, 1 point: drug predisposing to bleeding, 1 point. **** CKD was defined as eGFR < 60 mL/min/1.73 m^2^. ACEi = angiotensin converting enzyme inhibitor; AF = atrial fibrillation; ARB = angiotensin II receptor blocker; ASD = absolute standardized difference; BP = blood pressure; CCB = calcium channel blocker; CKD = chronic kidney disease; NOAC = non-vitamin K antagonist oral anticoagulant.

**Table 2 jcm-12-04579-t002:** Event rate in rate control group and rhythm control group before overlap weighting.

	Rate Control (N = 4506)			Rhythm Control (N = 2164)				
	Event, n	PYRs	Event/100 PYRs	Event, n	PYRs	Event/100 PYRs	HR (95% CI)	*p*-Value
Primary Outcome *	211	12,039	1.8	72	5124	1.4	0.73 (0.59~0.95)	0.022
Secondary outcome								
CV Death	20	12,185	0.2	5	5262	0.1	0.53 (0.20~1.42)	0.209
Stroke	81	12,087	0.7	20	5252	0.4	0.52 (0.32~0.85)	0.010
Acute coronary syndrome	22	12,165	0.2	4	5267	0.1	0.40 (0.14~1.16)	0.090
HF admission	16	12,180	0.1	7	5255	0.1	0.89 (0.37~2.18)	0.804

CI = confidence interval; CV = cardiovascular; HF = heart failure; HR = hazard ratio; PYRs = person-years. * The primary outcome was a composite endpoint of cardiovascular death, stroke, acute coronary syndrome, and heart failure admission event.

**Table 3 jcm-12-04579-t003:** Event rate in rate control group and rhythm control group after overlap weighting.

	Rate Control (N = 1293 [4506] *)			Rhythm Control (N = 1293 [2164] *)				
	Event, n	PYRs	Event/100 PYRs	Event, n	PYRs	Event/100 PYRs	Weighted HR (95% CI)	*p*-Value
Primary Outcome **	53	3448	1.5	45	3126	1.4	0.87 (0.66~1.17)	0.363
Secondary outcome								
CV Death	5	3485	0.1	3	3158	0.1	0.70 (0.25~1.99)	0.502
Stroke	20	3461	0.6	12	3150	0.4	0.60 (0.36~1.02)	0.060
Acute coronary syndrome	6	3481	0.2	2	3159	0.1	0.45 (0.15~1.36)	0.155
HF admission	4	3484	0.1	4	3153	0.1	0.93 (0.35~2.45)	0.882

CI = confidence interval; CV = cardiovascular; HF = heart failure; HR = hazard ratio; PYRs = person-years. * Weighted number of individuals [crude number of individuals]. ** The primary outcome was a composite endpoint of cardiovascular death, stroke, acute coronary syndrome, and heart failure admission event.

**Table 4 jcm-12-04579-t004:** Subgroup analysis of primary outcome in rate control group and rhythm control group after overlap weighting.

	Rate Control (N = 1293 [4506] *)			Rhythm Control (N = 1293 [2164] *)					
	Event, n	PYRs	Event/100 PYRs	Event, n	PYRs	Event/100 PYRs	Weighted HR (95% CI)	*p*-Value	P for Interaction
Sex									
Male (N = 1658)	32	2180	1.5	31	1987	1.6	1.02 (0.72~1.44)	0.913	0.190
Women (N = 928)	20	1268	1.6	13	1139	1.1	0.65 (0.39~1.09)	0.102	
Age									
Age < 75 (N = 1897)	29	2555	1.1	29	2034	1.4	1.05 (0.73~1.51)	0.801	0.150
Age ≥ 75 (N = 689)	24	893	2.7	15	822	1.8	0.67 (0.42~1.08)	0.098	
Onset of AF									
<3 month (N = 504)	9	548	1.6	10	546	1.8	1.09 (0.59~2.04)	0.784	0.430
≥3 month (N = 2082)	44	2900	1.5	35	2580	1.4	0.83 (0.60~1.14)	0.252	
CHA_2_DS_2_-VASc score									
≤2 (N = 1206)	15	1662	0.9	13	1481	0.9	0.87 (0.51~1.47)	0.602	0.933
≥3 (N = 1380)	38	1786	2.1	32	1645	2.0	0.89 (0.64~1.25)	0.507	
Previous stroke history									
Yes (N = 449)	15	561	2.7	13	559	2.3	0.87 (0.50~1.49)	0.602	0.964
No (N = 2137)	37	2886	1.3	31	2567	1.2	0.88 (0.63~1.23)	0.442	
HF history									
Yes (N = 294)	10	353	2.8	11	363	3.0	0.98 (0.54~1.80)	0.956	0.501
No (N = 2292)	43	3094	1.4	34	2763	1.2	0.83 (0.60~1.15)	0.259	
Left atrial diameter (LAD)									
LAD ≤ 4 cm (N = 640)	11	740	1.5	10	819	1.2	0.85 (0.46~1.58)	0.612	0.549
LAD > 4 cm (N = 1496)	35	1921	1.8	28	1677	1.7	0.90 (0.63~1.29)	0.566	
LAVI (mL/m^2^)									
LAVI ≤ 36 (N = 479)	9	649	1.4	6	549	1.1	0.80 (0.39~1.67)	0.553	0.876
LAVI > 36 (N = 1087)	29	1410	2.1	19	1181	1.6	0.79 (0.52~1.20)	0.274	
LVEF									
LVEF ≤ 40 (N = 117)	3	126	2.4	2	148	1.4	0.65 (0.18~2.43)	0.526	0.471
LVEF > 40 (N = 2058)	44	2589	1.7	37	2399	1.5	0.91 (0.66~1.24)	0.541	

AF = atrial fibrillation; CI = confidence interval; HF = heart failure; HR = hazard ratio; LAVI = left atrial volume index; LVEF = left ventricular ejection fraction; PYRs = person-years. * Weighted number of individuals [crude number of individuals].

## Data Availability

The data presented in the study are shared by researcher participating in this CODE-AF registry.

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
