# Peer review of "Rhythm Control and Cardiovascular or Cerebrovascular Outcomes in Patients with Atrial Fibrillation: A Study of the CODE-AF Registry"

_jcm, 2023, doi:10.3390/jcm12144579_

Round 1

Reviewer 1 Report

The paper by Choung et al. entitled "Rhythm control and cardiovascular or cerebrovascular outcomes in patients with atrial fibrillation: A study of the CODE-AF registry" addresses an interesting and topical issue in patients with atrial fibrillation: does maintaining rhythm have a prognostic effect?

Numerous studies have attempted to answer this question: Chung et al. analyzed data from a population collected in the CODE-AF registry to evaluate 2 different endpoints: the primary outcome being a composite of the rate of death due to cardiovascular causes, stroke, acute coronary syndrome, and heart failure and the secondary outcomes being individual components of the rate of death due to cardiovascular causes.

Analysis of the data shows that with regard to the primary outcome there is a significant difference in favor of patients in whom sinus rhythm is maintained compared with those in whom the rate-control strategy is chosen. But from the analysis of the weighted cumulative incidence curves for individual components of the primary composite outcome in rhythm- and rate-control groups for atrial fibrillation, it appears that for stroke only there is only a statistically significant difference in favor of those who are initiated to a rhythm-control strategy compared to those initiated to rate-control.

Chung's work is certainly interesting, but it suffers from the limitations of registry analysis, which are correctly stated in the "limitations" section.

The definition of "greater than or less than 3 months" allows this population to be compared with that of the EAST-AFNET-4 trial even though the differences with this trial are notable: overall, there were few patients undergoing ablation (309 overall, or 14.3% of the study population); moreover, it is unclear what elements led to the choice of an interventional versus ablation pathway (probably younger, with fewer comorbidities? Or were they CASTLE-AF like patients?): on all these points it is worthwhile for the authors to expand the comments in the discussion.

It is appropriate that the conclusion sentence in the abstract be rephrased so that it is more closely aligned with the results of the paper and that the advantages of the rhythm control strategy over the rate control strategy emerge more clearly.

Reviewer 2 Report

I read with great interest this well –written paper which examined a hotly debated topic in AF management. You can find my comments below.

Major Comments

1.       A graphical abstract is much anticipated to summarize and clarify your interesting results.

2.       Please provide definitions for the following conditions Line 95: paroxysmal, persistent and permanent AF

3.       Line 123: please clarify if NIH stroke scale was used. Furthermore did you include both disabling stroke and TIA ??

4.       Line 124: please provide 4th universal definition for MI diagnosis instead.

5.       Line 163: are there baseline echocardiographic data to include in a separate subgroup analysis?  For instance, atrial cardiomyopathy is involved in AF pathogenesis ( ESC 2021 guidelines) and should be investigated early at diagnosis (class II recommendation) . Furthermore, LAA indices are rigorously examined nowadays (see DOI: 10.1186/s43044-023-00356-3, DOI: 10.3389/fcvm.2022.971848, DOI: 10.1001/jamacardio.2022.5449, DOI: 10.1253/circrep.CR-23-0007).  It would be interesting to provide data regarding LA, LAA indices if available from your cohort.

6.       Table 1: define alcohol intake (e.g grams or drinks per day?) & CKD (e.g eGFR < 60??). Additionally, are there data on SGLT2 intake ?

Minor Comments

1.       Line 96-98: Did you consider adding HFmrEF (EF 40-50%)?

2.       Data on early rhythm control and potential benefits should be further emphasized (see/add this ref: DOI: 10.1016/j.jacc.2021.06.038)

3.       Some further refs to add: DOI: 10.1186/s43044-023-00356-3, DOI: 10.3389/fcvm.2022.971848, DOI: 10.1001/jamacardio.2022.5449, DOI: 10.1253/circrep.CR-23-0007, DOI: 10.1056/NEJMoa2029980, DOI: 10.1016/j.jacep.2018.07.007, DOI: 10.3390/diagnostics11091584)
